# Stimulus salience determines defensive behaviors elicited by aversively conditioned serial compound auditory stimuli

**Sarah Hersman[1], David Allen[1], Mariko Hashimoto[1], Salvador Ignacio Brito[1,2], Todd E Anthony[1,2,3]\***

[1]F.M. Kirby Neurobiology Center, Boston Children's Hospital and Harvard Medical School, Boston, United States; [2]Program in Neuroscience, Harvard Medical School, Boston, United States; [3]Departments of Psychiatry and Neurology, Boston Children's Hospital, Boston, United States

**Abstract** Assessing the imminence of threatening events using environmental cues enables proactive engagement of appropriate avoidance responses. The neural processes employed to anticipate event occurrence depend upon which cue properties are used to formulate predictions. In serial compound stimulus (SCS) conditioning in mice, repeated presentations of sequential tone (CS1) and white noise (CS2) auditory stimuli immediately prior to an aversive event (US) produces freezing and flight responses to CS1 and CS2, respectively (Fadok et al., 2017). Recent work reported that these responses reflect learned temporal relationships of CS1 and CS2 to the US (Dong et al., 2019). However, we find that frequency and sound pressure levels, not temporal proximity to the US, are the key factors underlying SCS-driven conditioned responses. Moreover, white noise elicits greater physiological and behavioral responses than tones even prior to conditioning. Thus, stimulus salience is the primary determinant of behavior in the SCS paradigm, and represents a potential confound in experiments utilizing multiple sensory stimuli.

**\*For correspondence:**
todd.anthony@childrens.harvard.edu

## Introduction

Learned temporal relationships between cue stimuli and aversive events allow individuals to avoid danger. For example, progressive darkening of clouds often precedes lightning storms, and dark skies prompt evacuation from exposed spaces. Other forms of threat prediction derive not from cue timing or sequencing but rather from the intensity or salience of a stimulus, such as an entrenched soldier who uses relative volume of auditory threat stimuli (e.g. foreign vehicles or voices) to gauge proximity of an advancing enemy and determine when to retreat. Human studies indicate that the specific neural circuits engaged during prediction of event occurrence depend on which cognitive strategy is used to solve a particular task (*Breska and Ivry, 2018*). Thus, determination of the neural mechanisms that regulate different forms of threat prediction, and the consequences when such mechanisms are dysfunctional, requires behavioral paradigms in which the cognitive processes engaged are clearly defined.

In SCS conditioning (*Fadok et al., 2017*), sequential presentation of two different auditory stimuli (pure tones followed by white noise, in that order) precedes delivery of an aversive unconditioned stimulus (US, footshock). Following repeated SCS-US presentations, mice exhibit distinct defensive behaviors to each SCS component: tones elicit freezing whereas white noise elicits flight. The paradigm thus appears to model natural behavioral shifts that occur as the perceived probability of directly encountering threat increases. As posited by 'predatory imminence theory', prey animals

**eLife digest** If you notice the skies above you becoming darker, your first thought might be to seek shelter. Experience will have taught you that darkening skies are often a sign of an approaching storm. Learning to recognise changes that occur prior to an unpleasant event can help us avoid danger. But this is not the only strategy people can use to predict when something bad is about to happen. Another option is to use the intensity, or salience, of sensory information. Soldiers fighting on the front line, for example, might rely on the loudness of enemy voices or vehicles to judge how close an advancing enemy is. This information will help them decide when to retreat.

Different brain processes are active when individuals use each of these two strategies to predict when an upcoming event will occur. One approach to study these processes is to use a technique called "SCS conditioning". This involves exposing mice to two sounds, followed by a mild electric shock administered to the feet. The first sound is a pure tone; the second is a burst of white noise. After repeated trials, mice begin to show distinct responses to the two sounds. They freeze in response to the tone but run away upon hearing the white noise.

These responses parallel behaviors seen in the wild. When mice detect a distant predator, they freeze to avoid detection. But if the predator comes too close for the mice to avoid being spotted, they instead try to flee. Some have argued that in the SCS task, mice learn that the white noise predicts an imminent shock. The mice therefore flee as soon as they hear it. By contrast, they learn that the tone predicts a delayed shock and therefore choose to freeze instead.

However, by tweaking the SCS procedure, Hersman et al. now show that even if the white noise occurs before the tone, it is still more likely than the tone to trigger an escape response. In fact, mice are more reactive to white noise than tones even if the sounds are never paired with shocks. This suggests that mice find white noise naturally more noticeable than tones. Moreover, Hersman et al. show that tones can also trigger escape responses if they are sufficiently intense. Together these results suggest that mice use the intensity of the stimuli – rather than the length of time between each stimulus and the shock – to decide whether to freeze or flee.

People with anxiety disorders often show exaggerated responses to things that do not pose a genuine threat. At present the pathways in the brain that are responsible for these excessive reactions are unclear. The results of Hersman et al. will aid research into the brain circuits that detect, assess and respond to threats. Understanding these circuits could in the future lead to better treatments for anxiety disorders.

---

initially freeze (to avoid detection) when predators are present at a distance, but then switch to flight (escape) to avoid entrapment if a predator becomes close enough that avoiding detection is no longer possible (*Blanchard and Blanchard, 1989*; *Blanchard et al., 1995*; *Bouton and Bolles, 1980*; *Fanselow, 1994*; *Fanselow and Lester, 1988*; *Perusini and Fanselow, 2015*).

Given the presence of a specific, repeating sequence of auditory stimuli preceding shock during conditioning, defensive behaviors elicited by individual components of the SCS could in principle be driven by learned CS-US temporal relationships. However, the form of both appetitive and aversive conditioned responses is known to vary substantially according to the particular properties of a given conditioned stimulus, even when the same underlying construct has been learned (*Holland, 1977*; *Holland, 1979*; *Holland, 1980*). Therefore, differences in the intrinsic properties of tone and white noise stimuli themselves, rather than their temporal relationship to the US, could underlie the distinct behaviors these stimuli evoke during SCS conditioning.

To define the key factors responsible for the topography of behavioral responding in this paradigm, we systematically varied CS-US temporal relationships and properties of SCS component stimuli during or following conditioning. We found that when presented at equal sound pressure levels (SPL), white noise elicits greater active defensive behavior than tones, irrespective of stimulus order during conditioning. Following standard tone-white noise SCS conditioning, each stimulus was also capable on its own of evoking either conditioned freezing or flight, according to SPL. Furthermore, when presented at equivalent SPL, white noise promoted greater arousal and simple locomotor responding than pure tone stimuli, even in the absence of any prior conditioning. Together, these

data argue that stimulus salience is the major factor determining the form of conditioned responses during SCS conditioning.

## Results

### White noise elicits active fear responses during SCS conditioning regardless of temporal relationship to the US

We first tested whether reversing the order of 7.5 kHz tone (TN) and white noise (WN) presentation during SCS conditioning reverses the behaviors these stimuli elicit. To distinguish responses due to learned CS-US associations from those due to sensitization or generalization, a control group was included with a 60 s 'gap' between the SCS and the US (Figure 1A–C). As evident from the motion traces (Figure 1D–F), all groups exhibited significantly greater motion during WN than TN, irrespective of the order that these stimuli were presented during training (Figure 1J–L). As conditioning progressed, mice in all groups began to exhibit active responses to the WN, including darting and jumping, behaviors quantified using an 'escape score' (Figure 1M–O, see Materials and methods).

Evidence that Pavlovian conditioning occurred to individual components of the SCS is as follows. First, freezing to the TN differed between paired Group 1 (G1) and gap Group 3 (G3) during conditioning. Freezing to TN was higher in G1 than G3 (3-Way ANOVA on Day 2, G1 vs. G3, Main Effect of Stimulus ($F_{(1,23)}$ = 429.5, p<0.0001), Main Effect of Trial ($F_{(4,92)}$ = 5.083, p=0.001), Group X Stimulus Interaction, ($F_{(1,23)}$ = 27.51, p<0.0001); Follow-Up Two-Way RM ANOVA for freezing just to the tone stimulus, Main Effect of Trial ($F_{(3.2, 75.9)}$=3.79, p<0.05), Main Effect of Group ($F_{(1,23)}$ = 6.41, p<0.05)). Second, a separate cohort of mice trained on the same protocol were tested for TN-elicited freezing in a novel context (Figure 1—figure supplement 1). Whereas mice in the gap group (G3 protocol) did not show significantly increased freezing between baseline and tone onset (p>0.05), mice in the paired group (G1 protocol) exhibited robust acute freezing upon tone onset (p<0.001)(Two-Way RM ANOVA, Main Effect of Stimulus ($F_{(1,13)}$ = 19.98, p<0.001), Stimulus X Group Interaction ($F_{(1,13)}$ = 5.492, p<0.05), Sidak's comparisons to determine which group drives the Main Effect of Stimulus). Third, motion and escape score during WN presentations differed between G1 and G3 during conditioning. Mice in G1 had higher motion to WN than G3 mice (3-Way ANOVA on Day 2, G1 vs. G3 Activity, Main Effect of Stimulus ($F_{(1,23)}$ = 69.89, p<0.0001), Main Effect of Group ($F_{(1,23)}$ = 11.75, p<0.01), Group X Stimulus Interaction ($F_{(1,23)}$ = 19.77, p<0.001); Follow-up Two-Way RM ANOVA for motion just to WN stimulus, Main Effect of Group ($F_{(1,23)}$ = 15.79, p<0.001)), and also had higher escape scores to the WN stimulus (3-Way ANOVA on Day 2, G1 vs. G3 Escape Score, Main Effect of Stimulus ($F_{(1,23)}$ = 67.85, p<0.0001), Main Effect of Group ($F_{(1,23)}$ = 15.98, p<0.001), Group X Stimulus Interaction ($F_{(1,23)}$ = 20.41, p<0.001); Follow-up Two-Way RM ANOVA for escape score just to WN, Main Effect of Group ($F_{(1,23)}$ = 18.25, p<0.001)). Although Group 2 (G2) did not show significantly different freezing, motion, or escape score compared to G3 (3-Way ANOVA on Day 2, G2 vs G3, no group differences or interactions for freezing, motion, or escape score), G2 did display differential behavior to the two CS stimuli across these same metrics and in the same direction as G1 (Figure 1E–N; 2-Way ANOVA with Trial and Stimulus as factors; details in Source Data).

Notably, G1 motion responses to WN on day 2 (Figure 1D) were largest immediately following stimulus onset and decreased thereafter until US exposure (paired t-test, average motion first two vs. last two seconds of CS2, trials 6, 7, p<0.01; trials 8, 9, p<0.05). Thus, imminence in the SCS paradigm does not appear to be determined by a cognitive process that uses cue order or hazard rate, and reversing stimulus order does not reverse behavior. Similar results were observed when these same experiments were performed with C57Bl/6J mice (Figure 1—figure supplement 2), the strain most comparable to that used in previous studies (Dong et al., 2019; Fadok et al., 2017). Together, these results suggest that threat prediction in the SCS paradigm may be related to intrinsic properties of the auditory stimuli themselves.

### White noise is inherently more arousing than 7.5 kHz tones in naïve, unconditioned mice

Mice can hear sounds from 1 kHz to 100 kHz, but sensitivity to specific frequencies varies dramatically over this range. For example, the minimal sound pressure levels (SPL) that mice can reliably

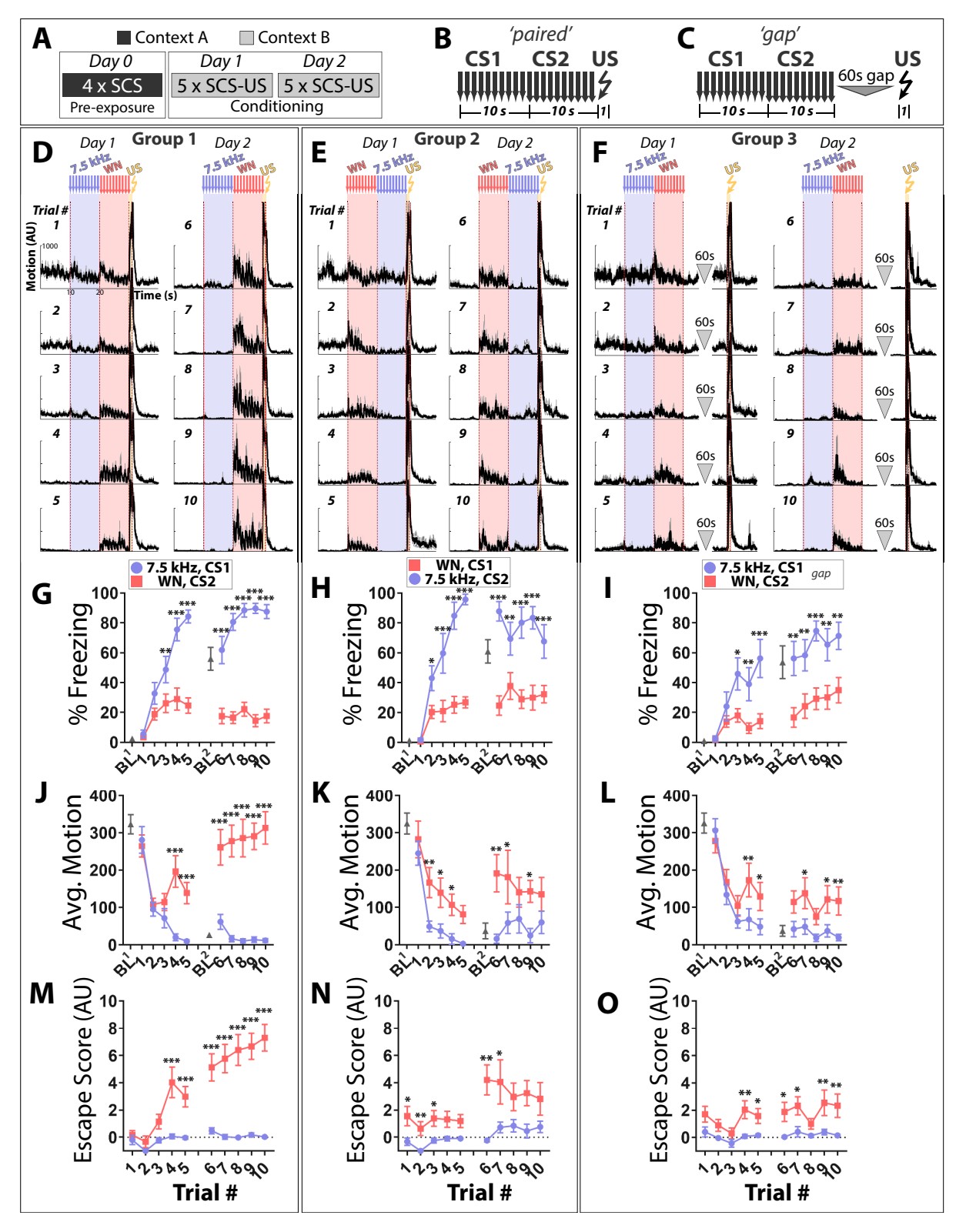

**Figure 1.** White noise elicits active fear responses during SCS conditioning regardless of temporal relationship to the US. (**A–C**) Protocol and structure of stimuli applied during conditioning for paired and unpaired groups. (**D–F**) Motion indices (mean ± SEM) showing movement in the absence or presence of stimuli (10 × 7.5 kHz pips, 75 dB, 0.5 s each at 1 Hz, blue; 10x white noise (WN) pips, 75 dB, 0.5 s each at 1 Hz, pink; 1 × 0.9 mA footshock, 1 s, yellow) for all 10 conditioning trials (Days 1 and 2). (**G–I**) Percentage time spent freezing during baseline (BL, 3 min prior to the first stimulus

*Figure 1 continued on next page*

*Figure 1 continued*

presented each day) and trials across each conditioning day. (**J–L**) Average motion during BL and trials. (**M–O**) Active fear behavior during each trial quantified as an escape score (see Materials and methods). CS order and pairing: group 1 (D,G,J,M: CS1 = 7.5 kHz, CS2 = WN; *n* = 15), group 2 (E,H,K, N: CS1 = WN, CS2 = 7.5 kHz; *n* = 10), and group 3 (F,I,L,O: CS1 = 7.5 kHz, CS2 = WN, gap; *n* = 10). Asterisks indicate significant difference between stimuli for a given trial (Two-way Repeated Measures ANOVA with Sidak multiple comparison test. Error bars indicate the SEM.

The online version of this article includes the following source data and figure supplement(s) for figure 1:

**Source data 1.** Raw data used to generate freezing, motion, and escape score plots and traces.
**Source data 2.** All statistical tests and significant comparisons with F and P values.
**Figure supplement 1.** Mice trained in a paired SCS protocol acutely freeze to tone presentation in a novel context, demonstrating Pavlovian Conditioning.
**Figure supplement 1—source data 1.** Raw data used to generate motion traces and freezing graph.
**Figure supplement 1—source data 2.** All statistical tests and significant comparisons with F and P values.
**Figure supplement 2.** Reversing order of tone and white noise presentation during SCS conditioning does not reverse the behaviors these stimuli elicit in C57Bl6/J mice.
**Figure supplement 2—source data 1.** Raw data used to generate freezing, velocity, escape score, and flight score plots and traces.
**Figure supplement 2—source data 2.** All statistical tests and significant comparisons with F and P values.

detect for 16 kHz tones is ~10 x lower (10 dB) than for 7.5 kHz tones (20 dB) (*Koay et al., 2002*). Given that the WN stimulus used here and previously (*Dong et al., 2019*; *Fadok et al., 2017*) is composed of frequencies between 1–20 kHz, one explanation for the above results is that WN stimuli are more efficiently detected and so of higher salience to mice than 7.5 kHz tones. To test this idea, we measured physiological and behavioral responses to unconditioned TN and WN stimuli from naive, head-fixed mice on running wheels that had not undergone conditioning of any kind or previously been exposed to these stimuli (*Figure 2A–C*). Surprisingly, we found that pupil dilation and simple locomotor responses on the running wheel were significantly greater to WN than TN (*Figure 2D–F*). In addition, comparison of the first three versus last three trials revealed that whereas TN responses habituate with repeated presentations, WN responses do not (*Figure 2G–N*). Thus, even in the absence of any association with an aversive US, TN and WN differ significantly in the magnitude of the physiological and behavioral responses they elicit.

## Stimulus intensity, not training order, determines the defensive behaviors elicited by SCS stimuli

This suggests that TN and WN are differentially salient to mice, which perceive the two stimuli as reflecting distinct points along the threat imminence continuum. A prediction of this model is that a 7.5 kHz CS presented at high SPL should be perceived as more imminent and elicit more escape than the exact same CS presented at low SPL. To test this, we performed a 'SPL step test' in which mice were presented with a SCS composed of two 7.5 kHz tones: CS1 is held constant at 75 dB while CS2 SPL magnitude begins at 55 dB and is stepped up by 5 dB each trial, finishing at 105 dB (*Figure 3A–C*). While predominantly freezing was observed at ≤85 dB, 7.5 kHz tones began to elicit escape behaviors in the paired group when CS2 ≥90 dB (*Figure 3D,F,H*). Further, escape scores for trials where CS2 ≥90 dB were significantly higher in group 1 (paired) than group 3 (gap; *Figure 3H, I*): 2-Way Repeated Measures ANOVA, Main Effect of Trial (F (4, 92)=3.208, p<0.05), Main Effect of Group (F (1, 23)=4.613, p<0.05). This argues that group one responses are at least in part influenced by perceived threat levels which are a function of conditioned fear, and are not simply a reflexive reaction to loud sounds. Moreover, escape at later trials was observed in response to CS2 but not CS1, demonstrating that these behavioral changes were not due solely to enhanced responsivity to any stimulus following repeated US exposure.

To determine whether behavioral responses to WN also scale with SPL, we performed a SPL step test using a simple WN CS presented in a novel context (*Figure 3J–L*). At low SPL (40–45 dB), WN elicited robust freezing and little to no escape behavior. In contrast, at higher SPL (≥60 dB), escape responses were common and freezing was minimal during WN presentations (*Figure 3M–O*). Thus, SCS fear conditioned TN and WN stimuli elicit freezing or flight behavior according to the SPL magnitude at which they are presented.

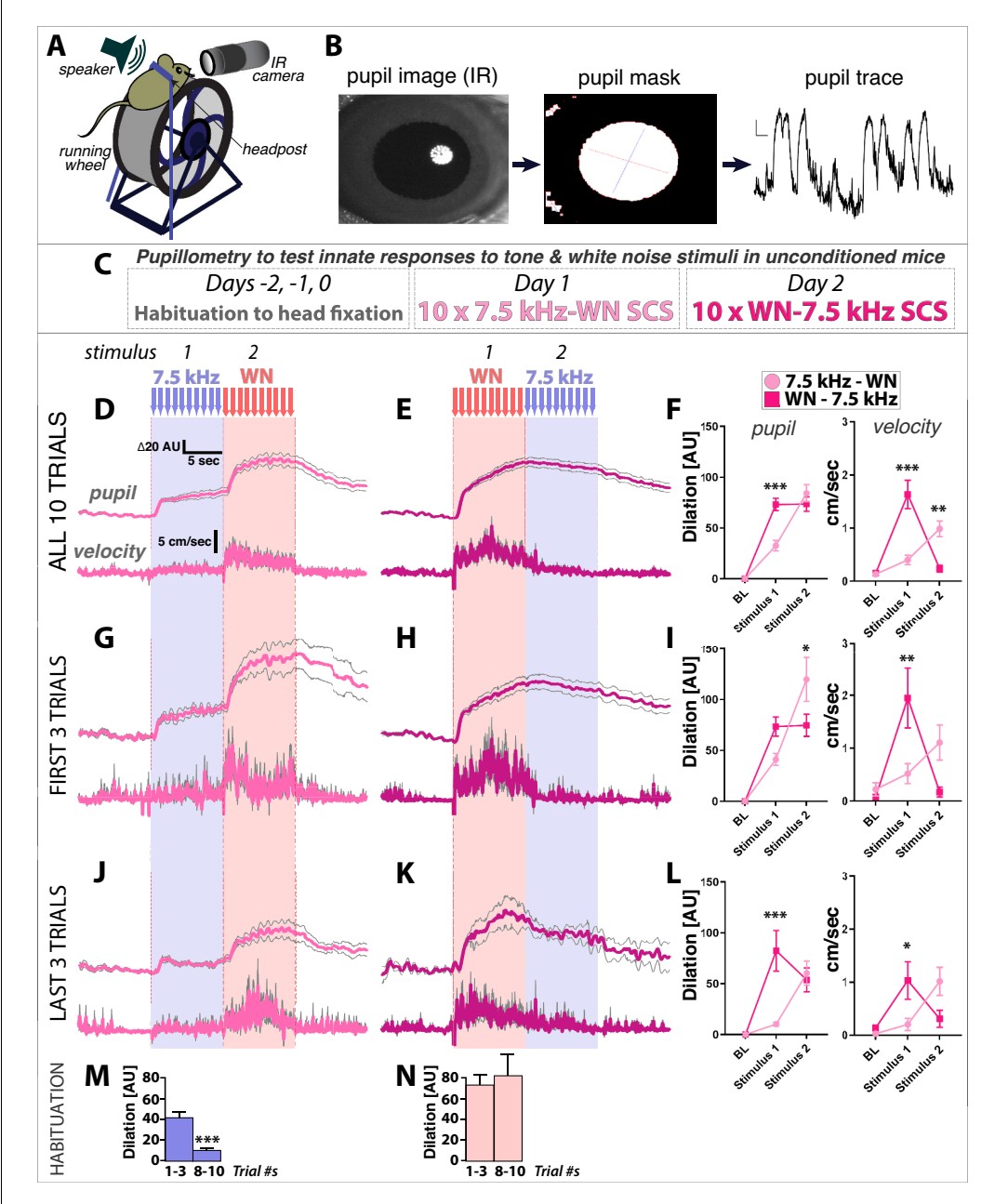

**Figure 2.** White noise is more innately arousing than 7.5 kHz tones in naïve, unconditioned mice. (**A–C**) Pupillometry setup, data processing, and protocol for measuring innate physiological responses to 7.5 kHz tone and white noise (WN) stimuli in unconditioned mice. (**D,E**) Pupil diameter relative to baseline (*top traces*) and running wheel velocity (*bottom traces*) are (**F**) both significantly greater in response to WN than tones. (**G–L**) Comparison of the first three (**G–I**) versus last three (**J–L**) trials reveals that responses to tones habituate more rapidly than to WN (*insets*, (**J,K**). Two-Way ANOVA (stimulus, group) with Sidak's multiple comparison tests (**F,I,L**) for pupil dilation relative to baseline (*left*) were done using mean values during the last 5 s of stimulus presentations (to account for slow kinetics of dilation), and for running wheel velocity (*right*) on mean values over the full 10 s stimulus presentations. (**M,N**) Comparison of first three versus last three trials of Tone response (**M**) shows habituation, while Noise response (**N**) does not. Traces and graphs plotted as mean ± SEM.

The online version of this article includes the following source data for figure 2:

**Source data 1.** Raw data used to generate pupil and velocity plots and traces.
**Source data 2.** All statistical tests and significant comparisons with F and P values.

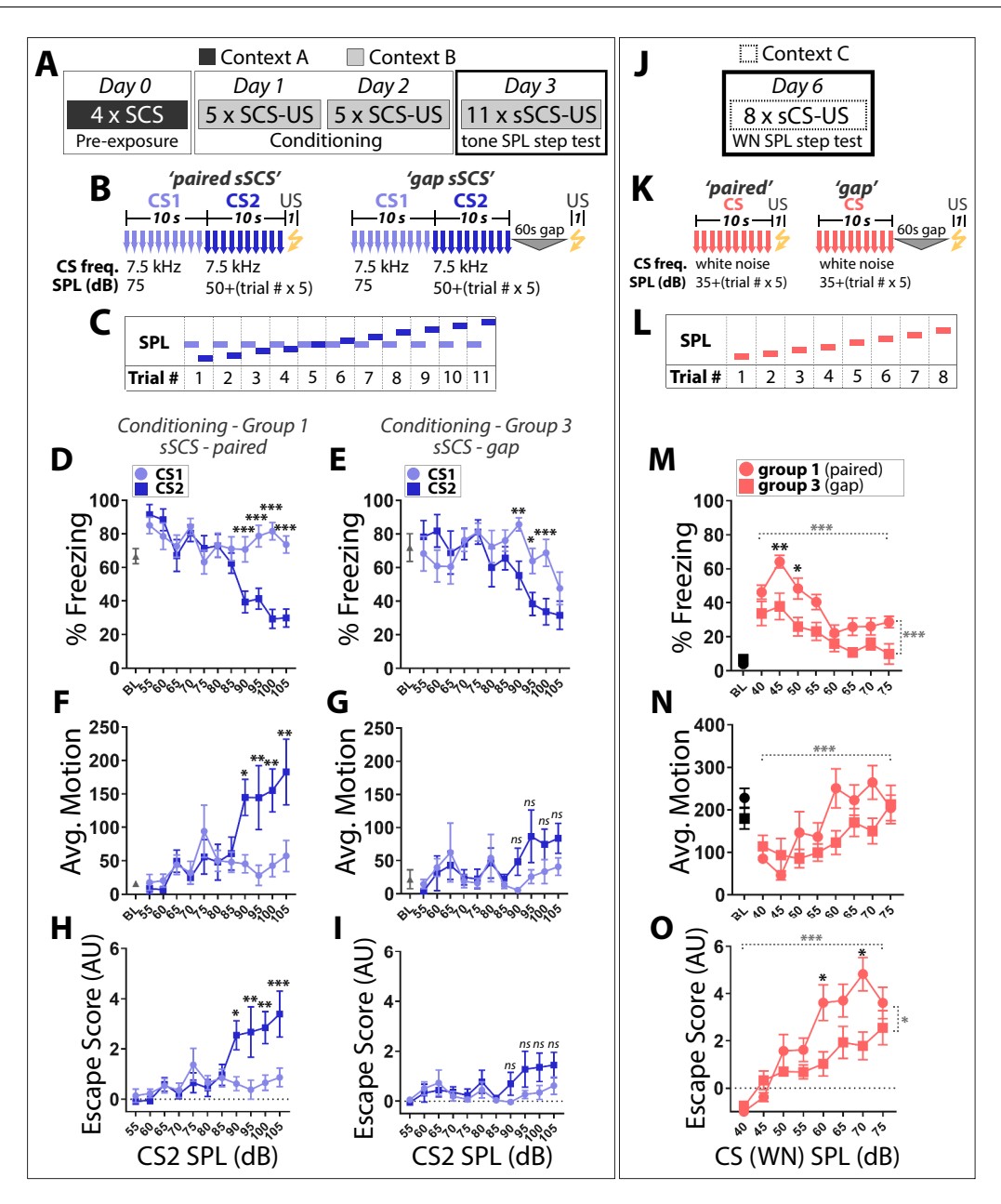

**Figure 3.** Stimulus intensity, not training order, determines the defensive behaviors elicited by SCS stimuli. (**A**) Mice conditioned in groups 1 and 3 (*Figure 1*) were run through a tone SPL step test on day 4. (**B**) The tone step SCS (sSCS) is composed of two 7.5 kHz tone stimuli in which CS1 is held constant at 75 dB while CS2 begins at 55 dB and is stepped up by 5 dB each trial. (**C**) Schematic of tone SPL step test. (**D,E**) Percentage time spent freezing. (**F,G**) Average motion. (**H,I**) Escape score. Paired sSCS (**D**, **F**, **H**); n = 15); gap sSCS (**E**, **G**, **I**); n = 10). Asterisks indicate significant difference between stimuli for a given trial. *ns*, not significant. (**J**) On day 6, a WN SPL step test was done in a novel context. (**K**) The WN step CS (sCS) is a white noise stimulus which begins at 40 dB and is stepped up by 5 dB each trial. (**L**) Schematic of WN SPL step test. (**M**) Percentage time spent freezing, (**N**) average motion, and (**O**) escape score. Paired sSCS (D,F,H); n = 15); gap sSCS (E,G,I; n = 10). Black asterisks indicate significant difference between groups for a given trial. Dashed horizontal gray brackets indicates significant main effect of SPL. Statistical comparisons were 2-Way Repeated Measures ANOVA, with Trial and Stimulus as factors. Dashed vertical gray brackets indicate significant main effect of group. Error bars indicate SEM. The online version of this article includes the following source data for figure 3:

**Source data 1.** Raw data used to generate freezing, motion, and escape score plots.
**Source data 2.** All statistical tests and significant comparisons with F and P values.

## Active fear behaviors are more potently elicited by 12 kHz than 3 kHz stimuli during pure tone SCS conditioning

Elicitation of robust escape by SCS conditioned 7.5 kHz tones required presentation at ≥90 dB, whereas both paired and unpaired mice began responding actively to WN stimuli at SPL as low as 50 dB. Although these stimuli differ in terms of frequency, they also differ with regards to signal regularity: whereas the 7.5 kHz tone is sinusoidal and periodic, WN is random and aperiodic. Therefore, although the above results could reflect differential sensitivity of mice to stimuli of different frequencies, they might alternatively be due to distinct defensive responses triggered by periodic versus aperiodic signals.

To test if frequency alone can influence defensive behaviors, we performed fear conditioning using a SCS composed of 3 and 12 kHz pure tones (*Figure 4*). These frequencies were chosen as: *a)* the threshold SPL in mice is ~100 x lower for 12 kHz than 3 kHz pure tones (*Koay et al., 2002*); perceived loudness of these two stimuli should thus differ when presented at standard SPL used during conditioning, similar to a 7.5 kHz/WN SCS; and *b)* 12 kHz is well separated from 17 to 20 kHz, a range that may be innately aversive in mice (*Beckett et al., 1996*; *Blanchard et al., 1992*; *Cuomo et al., 1992*; *Evans et al., 2018*; *Mongeau et al., 2003*). As conditioning progressed, paired groups exhibited higher motion, less freezing, and more escape to the 12 kHz than 3 kHz CS, regardless of the order in which the stimuli were presented during training (*Figure 4E–M*). Thus, despite having no apparent intrinsic aversive valence, 12 kHz tones can elicit greater active threat responses than 3 kHz tones presented at equivalent SPL during SCS conditioning.

Freezing and escape behavior in this 'two-tone' SCS protocol resulted from Pavlovian Conditioning. Though groups did not differ in freezing behavior to the 3 kHz tone during conditioning, this difference was revealed in a novel context tone test (*Figure 4N–P*). Elevated motion and escape behaviors to the 12 kHz tone occurred only in the paired groups, indicating that these behaviors are conditioned responses (2-Way RM ANOVA, G4 vs. G6, Motion to 12 kHz Tone: Main Effect of Trial ($F_{(2.5, 45.7)}$=3.31, $p<0.05$), Main Effect of Group ($F_{(1,18)} = 8.64$, $p<0.01$); 2-Way RM ANOVA, G4 vs. G6, Escape score to higher Tone: Main Effect of Trial ($F_{(2.7, 48.0)}$=4.36, $p<0.05$), Main Effect of Group ($F_{(1,18)} = 10.1$, $p<0.01$); 2-Way RM ANOVA, G5 vs G6, Escape score to 12 kHz Tone: Main Effect of Group ($F_{(1,18)} = 4.49$, $p<0.05$). As observed for the TN and WN stimuli (*Figure 1*), reversing stimulus order reduced the magnitude of the elevated activity and escape to the high-salience stimulus, but did not reverse the behaviors elicited by the two stimuli.

## Discussion

In conclusion, we found that audio frequency properties strongly influence the defensive behaviors elicited by SCS fear conditioned auditory stimuli. Escape behaviors were most potently triggered by stimuli that contain frequencies to which mouse hearing is most sensitive, an effect that was independent of the order in which auditory stimuli were presented during learning. In addition, pure tones that elicit freezing at typical experimental sound pressure levels can promote conditioned escape when presented at higher levels. These data argue that stimulus salience, not temporal proximity to the US, is the primary means by which mice assess imminence and engage appropriate defensive strategies in the SCS paradigm. This would appear to be similar mechanistically to how mice respond to innately threatening visual stimuli, where the probability and intensity of escape behaviors scale with visual stimulus salience (*Evans et al., 2018*).

An implication of this work is the critical need to consider the behavioral sensitivity of experimental subjects to auditory stimuli of different frequencies. Psychophysical studies have demonstrated that all species have a particular range of frequencies that they hear well (i.e. which are audible at 10 dB); stimuli outside of this range may need to presented at substantially higher SPL in order to be efficiently detected. In addition, although most laboratory animals exhibit overlap in their hearing ranges, there can be significant differences in their sensitivity to particular frequencies, even among closely related species. For example, whereas the 10 dB threshold includes frequencies ranging from ~5–40 kHz in rats, this range is very narrow in mice and limited to frequencies close to 16 kHz (*Heffner and Heffner, 2007*). Differences can also exist across mouse strains and between different ages of the same strain. For example, C57BL/6J mice undergo hearing-loss induced plasticity that by 5 months of age results in loss of responsivity to high frequency tones (>20 kHz) with concomitantly enhanced behavioral sensitivity to middle (12–16 kHz) but not low (4–8 kHz) frequency stimuli

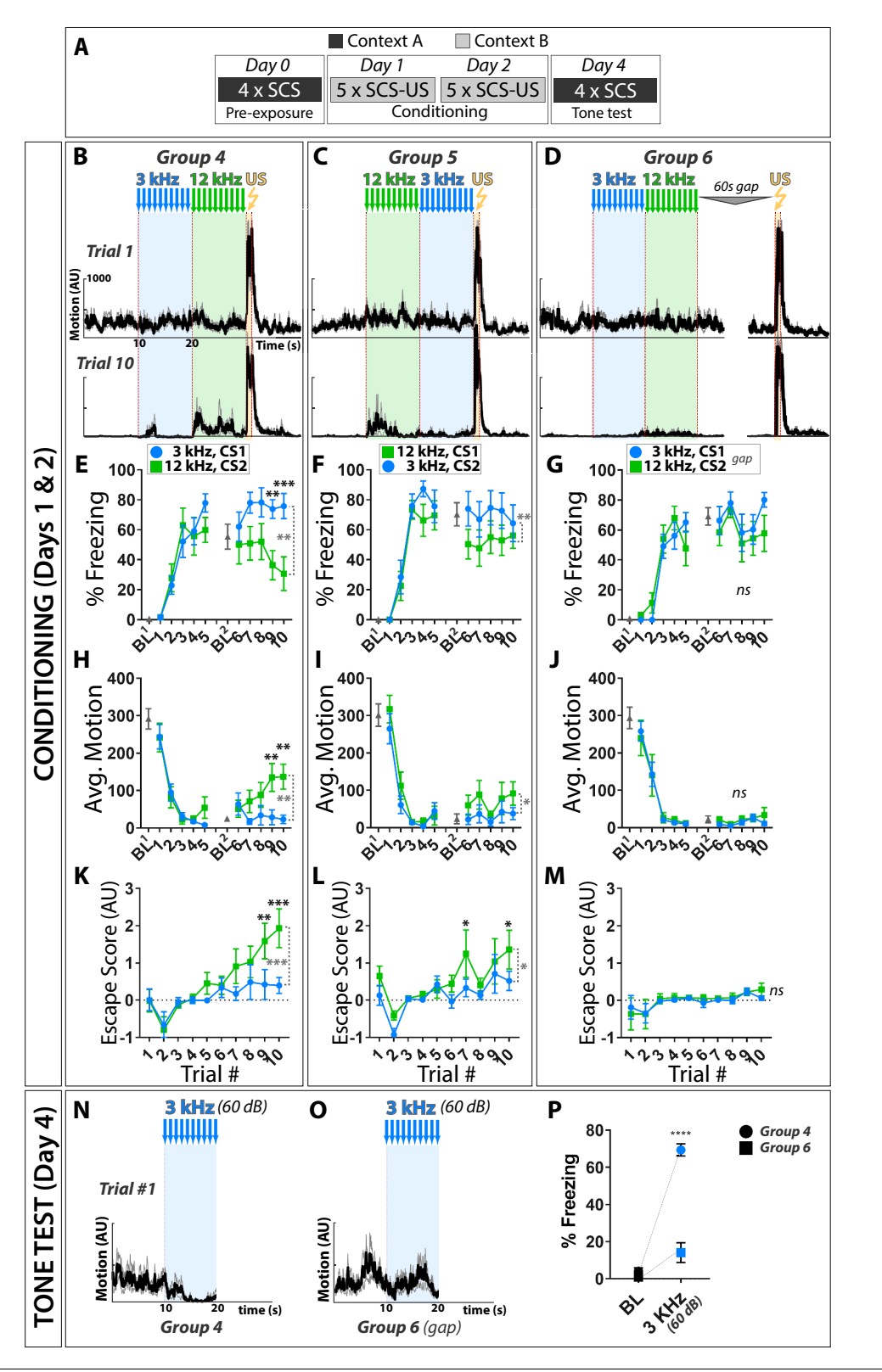

**Figure 4.** Active fear behaviors are more potently elicited by 12 kHz than 3 kHz stimuli during pure tone SCS conditioning. (A) Contribution of CS audio frequency, order presented relative to US, and pairing were assessed by conditioning with a SCS composed of 3 and 12 kHz pure tones; conditioning was done as in *Figure 1A*. (B–D) Motion indices (mean ± SEM) show locomotor responses to stimuli (3 kHz pips, blue; 12 kHz pips, green; 0.9 mA footshock, yellow) for trials 1 (Day 1, *top*) and 10 (Day 2, *bottom*). (E–G) Percent time spent freezing. (H–J) Average motion. (K–M) Escape score. CS

*Figure 4 continued on next page*

*Figure 4 continued*

order and pairing: group 4 (B,E,H,K: CS1 = 3 kHz, CS2 = 12 kHz; *n* = 10), group 5 (C,F,I,L: CS1 = 12 kHz, CS2 = 3 kHz; *n* = 10), and group 3 (D,G,J,M: CS1 = 3 kHz, CS2 = 12 kHz, unpaired; *n* = 10). (**N–P**) Tone tests established that lack of active responses to the 75 dB 3 kHz CS was not due to it being inaudible, as paired mice exhibited robust freezing to this stimulus when presented at an even lower SPL (60 dB) outside of the conditioning context. Black asterisks indicate significant difference between groups for a given trial. Dashed vertical gray brackets indicate significant main effect of CS type. Error bars indicate the SEM.

The online version of this article includes the following source data for figure 4:

**Source data 1.** Raw data used to generate freezing, motion, escape plots and traces.
**Source data 2.** All statistical tests and significant comparisons with F and P values.

(*Carlson and Willott, 1996*; *Willott et al., 1994*). Moreover, certain frequencies may be innately aversive in rodents: rats emit and respond defensively to alarm vocalizations near 20 kHz (*Beckett et al., 1996*; *Blanchard et al., 1992*; *Cuomo et al., 1992*), and 17–20 kHz ultrasonic sweeps can elicit robust freezing and flight behaviors in mice (*Evans et al., 2018*; *Mongeau et al., 2003*). White noise stimuli, which are both aperiodic and include 17–20 kHz frequencies, may thus be uniquely salient to mice under conditions of impending potential threats due to recruitment of dedicated defensive circuits tuned to innately threatening auditory stimuli. Indeed, in conventional fear conditioning to a simple CS composed of a single auditory stimulus, significantly more flight behavior was evoked by a white noise CS than a tone CS (*Fadok et al., 2017*).

Importantly, discrimination studies that employ multiple auditory cues could be complicated both by variations in the ability of subjects to perceive different frequencies as well as potential innate valence associated with certain stimuli. For example, aversively conditioning a high intensity US with a 5 kHz CS+ followed by a generalization test using a higher salience CS- such as white noise could yield misleading conclusions if subjects exhibit escape behaviors to the CS- and, as is common, freezing is the only metric used to assess cue responsivity. Such confounds may be best avoided by assaying discrimination using tasks which measure behavioral responses to distinct patterns of a single, constant intensity sensory stimulus (e.g. drifting visual gratings of different orientation [*Burgess et al., 2016*]). Interpretation of discrimination studies that employ auditory stimuli would benefit from counterbalancing assignment of CS+ and CS- stimuli (*Sanford et al., 2017*), and also from use of stimuli at frequencies and SPL that are detectable but do not trigger active fear behaviors.

Severe stress can result in persistent generalization or sensitization of threat responding, such that stimuli which normally elicit little to no response come to evoke robust defensive behaviors. For example, in the stress-enhanced fear learning (SEFL) model, exposure to inescapable shocks in the absence of auditory stimuli results in nonassociative freezing to white noise in a novel context on the following day (*Perusini et al., 2016*). Although we observed white noise-elicited escape behavior in group 3 ('gap', *Figure 1F and O*), these responses cannot be attributed directly to generalization as it remains possible that some CS-US association formed despite the 60 s gap (i.e. via trace conditioning). Thus, the extent to which white noise can nonassociatively elicit active defensive behavior will need to be determined in future experiments where training is performed with a US presented in the complete absence of CS stimuli, as done in the SEFL model.

Previous work provided behavioral and neurophysiological evidence that SCS fear conditioned tone and white noise stimuli acutely elicit distinct defensive states indicative of different points along the threat imminence continuum (*Fadok et al., 2017*). We have found that these defensive states track with the frequency and intensity of the conditioned stimuli, not order of CS presentation during learning. This argues that threat imminence in this model is determined primarily via the salience of threat-predictive auditory stimuli which, together with recent experience (*Mongeau et al., 2003*) and current fear levels, determines the threshold for switching from freezing to flight. Our results contrast with those of another study (*Dong et al., 2019*), which reported that training with a 'reversed SCS' (white noise-tone-US) reverses the behaviors elicited (i.e. mice freeze to the WN but exhibit flight to the tones). As the experimental procedures used in both studies were essentially the same, the explanation for the discrepant results is presently unclear. However, while the B6J mice used here were obtained from JAX, the mice used in Dong et al. were of undefined substrain ('C57Bl/6') and obtained from a different vendor. Therefore, it is possible that different mouse

strains utilize distinct neural processes to assess threat imminence. Future work will be required to determine if this is indeed the case and if so, the mechanistic underpinnings of such differences.

Although reversing the order of the white noise and tone stimuli during training did not qualitatively alter the type of behaviors elicited by the CSs, this switch did have a quantitative effect. Specifically, white noise elicited significantly less escape behavior when it preceded rather than followed the tone during training (*Figure 1*). One potential explanation for this result is that compound stimuli which increase in salience from CS1 to CS2 are more naturalistic and produce higher arousal levels and greater learning than the reverse order. Indeed, tonal stimuli which sweep from low up to high frequencies are rated by human observers as more alarming than high to low sweeps (*Catchpole et al., 2004*). Similarly, frequency upsweeps are associated with elevation of attention and arousal, whereas downsweeps are thought to have a calming effect (*Owren and Rendall, 2001*). Use of compound stimuli that either increase or decrease in salience from CS1 to CS2 might thus have opposing influences on arousal, resulting in either optimal or suboptimal states for sensory signal processing and learning (*Aston-Jones and Cohen, 2005*; *McGinley et al., 2015*; *Yerkes and Dodson, 1908*).

Finally, we note that conditioned responses exhibited at the onset of a CS can differ qualitatively from those displayed near CS offset (*Holland, 1980*). It thus remains possible that temporal factors make some contribution to defensive responding in SCS conditioning. Given the potent influence of stimulus salience, resolution of this issue will likely require the use of a SCS comprised of distinct component stimuli that can be clearly discriminated and yet are also matched for salience.

# Materials and methods

### Key resources table

| Reagent type (species) or resource | Designation | Source or reference | Identifiers | Additional information |
|---|---|---|---|---|
| Strain, strain background (*Mus musculus*) | FVBB6 F1 | Taconic stock #*FVB/Ntac* x Envigo stock #C57Bl/6NHsd | RRID:MGI:5653121 RRID:MGI:5658877 | cross of FVB/N to B6N |
| Strain, strain background (*Mus musculus*) | C57BL/6J | The Jackson Laboratory (Bar Harbor, ME) | RRID:IMSR_JAX:000664 | JAX stock # 000664 |
| Software, algorithm | SPSS | IBM | RRID:SCR_002865 | |
| Software, algorithm | Bonsai | Open Ephys | RRID:SCR_017218 | Version 2.3 |
| Software, algorithm | MATLAB | Mathworks (Natick, MA) | RRID:SCR_001622 | Version R2017b |

## Subjects

Male FVBB6 F1 hybrid mice (3–5 months of age, 25–30 g) were used for all experiments except those in *Figure 1—figure supplement 1*, which used C57Bl/6J mice (JAX). All mice were singly housed beginning one week prior to and throughout training and testing, and maintained on a 12 hr reverse light/dark cycle with access to food and water ad libitum. All behavioral tests were conducted during the dark phase, beginning not before one hour of lights OFF and ending not later than one hour before lights ON. Animals were randomly assigned to the experimental groups. The behavioral procedures used in this study were approved by the Institutional Animal Care and Use Committee at Boston Children's Hospital.

## Apparatus

Behavioral training used fear conditioning chambers (30 × 25×25 cm, Med-Associates, Inc St. Albans, VT), equipped with a Med-Associates VideoFreeze system. The boxes were enclosed in larger sound-attenuating chambers. Aspects of the boxes were varied to create two distinct contexts. The pre-exposure and testing context were composed of a white Plexiglas floor insert and a curved white Plexiglas wall insert with a hole over the wall speaker, making the rear walls of the

chamber into a semi-circle. The ceiling and front door were composed of clear Plexiglas. The overhead light was off and the box was cleaned with 1% acetic acid. The conditioning context was comprised of a rectangular chamber with aluminum sidewalls and a white Plexiglas rear wall. The grid floor consisted of 16 stainless steel rods (4.8 mm thick) spaced 1.6 cm apart (center to center). Pans underlying each box were sprayed and cleaned between mice. Fans mounted above each chamber provided background noise (65 dB). The experimental room was brightly lit with an overhead white light. Animals were kept in a holding room and individually transported to the experimental room in their home cage. Chambers were cleaned with soap and water following each day of behavioral testing.

### Serial compound stimulus (SCS) fear conditioning

For tone-white noise SCS, three groups of mice were conditioned with compound stimuli consisting of ten pure tone pips (7.5 KHz, 75 dB, 0.5 s duration at 1 Hz), ten white noise pips (WN, 75 dB, 0.5 s duration at 1 Hz), and a foot shock (0.9mA, 1 s duration). The order and pairing differed between groups: Group one received Tone-WN paired with shock, Group two received WN-Tone paired with shock, and Group three received Tone-WN not directly paired with shock (i.e. 60 s gap in between CS2 and US). All groups had a 3 min baseline period prior to the first CS and 30 s after the final shock. Groups 1 and 2 had a 60 s average pseudorandom ITI (range 50–90 s), while Group 3 had a 180 s average pseudorandom ITI (range 150–200). For pure tone SCS conditioning, the protocols were the same except that the tone and white noise stimuli were replaced with two pure tone stimuli: 3 KHz (75 dB, 10 × 0.5 s duration pips at 1 Hz) and 12 KHz (75 dB, 10 × 0.5 s duration pips at 1 Hz). On the day 0 of both experiments, mice were placed into the pre-exposure context and received four CS-alone trials. On Days 1 and 2, mice were placed into the conditioning context, where they received five CS trials that included shock. SPL step tests were run as indicated in the figures.

### Quantification of behavior

Freezing behavior, average motion, and maximum motion were calculated using motion indices determined using automated near infrared (NIR) video tracking equipment and computer software (VideoFreeze, Med-Associates Inc), as previously described (*Zelikowsky et al., 2013*). Escape behaviors were scored manually from video files to count the number of darts and jumps. Darts were defined as rapid crossings preceded by immobility; jumps were defined as rapid movements in which all four paws left the floor. These behaviors were summed to determine the number of escape behaviors per mouse per trial, and used to quantify the vigor of responses to particular auditory stimuli via an 'escape score'. As most mice were freezing throughout baseline (BL) periods on conditioning day 2 (resulting in a motion index = 0), computation of a 'flight score' which compares motion during CS presentation versus BL as a CS/BL *ratio* (similar to what was done previously using velocity [*Fadok et al., 2017*]) was problematic due to most ratios having 0 in the denominator. We therefore calculated an 'escape score' by taking the *difference* in average motion index (MI) during CS versus the baseline for each trial (i.e. the 10 s period preceding delivery of a CS), dividing this by 100, and then adding one point for each dart or two points for each jump observed during that particular stimulus and trial: escape score = $(MI_{CS} – MI_{BL})/100 + 1$ (for each dart) + 2 (for each jump).

### Pupillometry

Mice with stainless steel head posts were head-fixed on a running wheel, and pupils illuminated with an infrared LED and imaging with a FLIR Flea3 USB 3.0 camera at 30fps. Importantly, mice used for these experiments had not previously received any type of conditioning nor been exposed to either tone or white noise stimuli. To extract pupil diameter traces, the pupil was thresholded and binarized in Bonsai 2.3 using a custom workflow (OpenCV). The resulting image was dilated and eroded to remove noise from the pupil edge, and the largest radius of the oval is extracted as pupil diameter. Blinks were removed in MATLAB. Following habituation to head-fixation on the wheel for three days (10 min per day), mice were exposed to ten trials of the Tone-WN stimuli alone; the following day they received ten trials of the WN-Tone stimuli alone. To minimize the influence of 'ceiling effects', trials were excluded when pupil diameter exceeded the mouse's own 50th percentile in the 5 s prior to stimulus onset. All velocity traces were included.

## Statistical analysis

Data were analyzed with t-tests or two-way repeated-measures ANOVAs, with Sidak post hoc analysis correcting for multiple comparisons where appropriate. Sample size was pre-determined from previously published research and from pilot experiments performed in the laboratory. Experiments in *Figure 1* were replicated two (groups 2 and 3) or three (group 1) times using separate cohorts of animals. Experiments in *Figure 2* were replicated twice using separate groups of animals. Experiments in *Figures 3* and *4* were performed once. Experiments in *Figure 1—figure supplement 1* were performed once. Experiments in *Figure 1—figure supplement 2* were replicated twice with separate cohorts of animals. In all instances, these were 'biological replicates' (i.e. different mice for each experiment). Lab personnel were blind to experimental group during scoring. Statistical significance is labeled as $*p<0.05$, $**p<0.01$, and $***p<0.001$.

## Acknowledgements

We thank Delaney Foley for running the initial SCS conditioning experiments, and members of the Andermann lab for helpful discussions. This work was supported by NIH training grant #T32 NS007473 and Hearst Fellowship (SH), and grants from the National Institute of Mental Health (#1R01MH117421-01A1), Whitehall Foundation, Charles Hood Foundation, Tommy Fuss Center for Neuropsychiatric Disease Research, Harvard Neurodiscovery Center, Harvard University Milton Fund, and Harvard Brain Initiative (TEA).

## Additional information

### Funding

| Funder | Grant reference number | Author |
| --- | --- | --- |
| National Institutes of Health | 1R01MH117421-01A1 | Todd E Anthony |
| National Institutes of Health | T32 NS007473 | Sarah Hersman |
| Whitehall Foundation | 2016-05-99 | Todd E Anthony |
| Charles H. Hood Foundation | 2017-10-1 | Todd E Anthony |
| Boston Children's Hospital | Tommy Fuss Center for Neuropsychiatric Disease Research | Todd E Anthony |
| Harvard NeuroDiscovery Center | | Todd E Anthony |
| Harvard University | Milton Fund | Todd E Anthony |
| Harvard Medical School | Harvard Brain Science Initiative | Todd E Anthony |

The funders had no role in study design, data collection and interpretation, or the decision to submit the work for publication.

### Author contributions

Sarah Hersman, Formal analysis, Investigation, Writing - original draft; David Allen, Software, Investigation, Methodology; Mariko Hashimoto, Salvador Ignacio Brito, Investigation; Todd E Anthony, Conceptualization, Formal analysis, Writing - original draft, Writing - review and editing

### Author ORCIDs

Todd E Anthony https://orcid.org/0000-0002-7284-7556

### Ethics

Animal experimentation: The behavioral procedures used in this study were performed in strict accordance with the recommendations in the Guide for the Care and Use of Laboratory Animals of the National Institutes of Health. All animals were handled according to protocols approved by the

institutional animal care and use committee (IACUC) at Boston Children's Hospital (Protocol 18-07-3726R).

### Decision letter and Author response
Decision letter https://doi.org/10.7554/eLife.53803.sa1
Author response https://doi.org/10.7554/eLife.53803.sa2

## Additional files

### Supplementary files
• Transparent reporting form

### Data availability
All data generated or analysed during this study are included in the manuscript and supporting files. Source data files have been provided for all figures in MS Excel format, with primary measurements in one file and statistical analyses in another file.

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
