## [Decision Letter]

**Acceptance summary:**

In this paper, the authors test assumptions about the basis of differential responding in a serial fear conditioning preparation, in which rodents are exposed to a tone->white noise->shock US. Normally rodents react to this by freezing to the tone and then exhibiting flight behavior to the white noise. Previous work has assumed that the changeover in behavior from freezing to flight is due to the temporal relationship to the US, with distal behaviors being directed at avoiding detection by a predator and proximal behaviors being directed at escape. The current study challenges this assumption, and instead shows that a substantial component of the changeover in behavior is instead driven by the inherently higher salience of the white noise that is usually used for the proximal cue. The reviewers are in agreement that this is an important study with regard to serial fear conditioning, and that it further makes an important general point for those of us interested in the intersection between behavior and neuroscience, which is that there are many factors that control the form of any behavioral response.

**Decision letter after peer review:**

Thank you for submitting your article "Stimulus Salience Determines Defensive Behaviors elicited by aversively conditioned serial compound auditory stimuli" for consideration by *eLife*. Your article has been reviewed by three peer reviewers, including Geoffrey Schoenbaum as the Reviewing Editor and Reviewer #1, and the evaluation has been Laura Colgin as the Senior Editor. The following individuals involved in review of your submission have agreed to reveal their identity: Gavan McNally (Reviewer #2).

The reviewers have discussed the reviews with one another and the Reviewing Editor has drafted this decision to help you prepare a revised submission.

Essential revisions:

The reviewers are in agreement that this is an important study with regard to serial fear conditioning, and that it further makes an important general point for those of us interested in the intersection between behavior and neuroscience, which is that there are many factors that control the form of any behavioral response. For this reason, counterbalancing and other procedures are very important. The current study is a case study of this. In our discussion, all three reviewers generally agreed that essential revisions could be dealt with in the text. All three appreciated the author's point regarding salience but felt that it would be better to acknowledge that many factors determine the "topography" of the response, instead of concluding that salience is the entire cause in all prior studies. As part of this, it was felt that putting this result in a larger context regarding the form of a behavior would be good. Peter Holland's work is particularly relevant, but there are many other examples. Possibly if the authors are unsure about this, they could let us know. There were also important concerns about the statistics raised by R2 (3-factor anova) and questions regarding whether longer intervals were used to rule out nonassociative responding, which are important to address in the revision. The reviews are included in their entirety below.

*Reviewer #1:*

In this paper, the authors test assumptions about the basis of differential responding in a serial fear conditioning preparation, in which rodents are exposed to a tone->white noise->shock US. Normally rodents react to this by freezing to the tone and then exhibiting flight behavior to the white noise. Previous work has assumed that the changeover in behavior from freezing to flight is due to the temporal relationship to the US, with distal behaviors being directed at avoiding detection by a predator and proximal behaviors being directed at escape. The current study challenges this assumption, and instead shows that a substantial component of the changeover in behavior is instead driven by the inherently higher salience of the white noise that is usually used for the proximal cue. Overall, the experiments are well done and provide convincing evidence supporting what is essentially a cautionary tale regarding the importance of careful, well-controlled behavioral designs. My criticism are entirely suggestions for caveating or softening the conclusions a bit.

There are two main areas I think could be made clearer and perhaps softened a bit. The first concerns the relationship between these findings and the general idea that behaviors to cues can differ based on temporal factors, salience, or even cue modality. I do not know the SCS field particularly, but even without a compound cue, it is well documented that both unconditioned and conditioned behaviors differ across time, particularly during a long (10, 20, 30s) cue. These behaviors and their relationships can also differ based on the amount of training, and the density of reward I believe. So, it seems to me, that there are many factors that can explain differential responding of the sort that seems to be dogmatically-highlighted in the SCS literature. I wonder if a more general introduction that acknowledges some of this complexity might be considered, versus what seems to be a dichotomy in the current introduction?

Related to this, I think the authors should soften their conclusions a bit. At present, it seems to me that they are saying the SCS effect is solely due to salience differences. While this may be the case, I think this is conclusion goes beyond what is necessary and the data. At best, what the authors demonstrate is that your salience can play a major role in determining the behavior. But I think this is not the same as staying the temporal relationship to the US is not important or never plays a role. Indeed, logic suggests, as the authors Introduction points out, that it surely should. The question just is what training procedures need to be used to demonstrate this conclusively – i.e. excluding salience and other factors. This I think is one of the main messages of this very nice study – that it is important to consider and control for these effects carefully. I think if the authors can make these points in a bit more nuanced way, it will improve the impact of the study.

*Reviewer #2:*

In an interesting series of experiments, Fadok et al., (2017) and Dong et al., (2019), reported that mice would show flight responses to auditory CSs that signalled an imminent shock US. These papers used a serial compound conditioning procedure whereby a tone CS was presented then a white noise CS then shock. Serial compound conditioning is, of course, a very old procedure. The novelty in these papers was the finding that mice would freeze to the distal tone CS then engage in active defense (escape, movement) to the proximal white noise CS. These effects were interpreted in terms a shift from passive to active defense as US imminence increased and were consistent with the important and influential predatory imminence models of Fanselow. Although Fadok et al., did not counterbalance the identity of the CSs (i.e. they used tone then white noise), Dong et al., showed the same effect with white noise then tone presentations (freezing to the white noise and flight the white noise). Fadok et al., also showed that the white noise itself was not aversive and did not elicit escape responses or flight behaviour in the absence of footshock (Extended data Figure 1H-J).

In the present manuscript, Hersman et al. provide a careful behavioral analysis of the topography of unconditioned and conditioned responses to white noise and tone CSs paired with shock. Their main claim is that physical properties of the CSs (frequency and sound pressure levels), not their temporal imminence, determines the topography of responding. These are very interesting experiments addressing an important question. On the one hand, if the field simply used appropriate counterbalancing of the identity of CSs in individual papers, we may not be having these discussions. On the other hand, the manuscript is a systematic investigation into the effects of auditory CS properties on the topography of defensive behaviour. The manuscript is well written, economical, and well presented.

I had the following three comments on the designs, analyses, and interpretation:

1) It is well established that multiple factors determine the topography of behaviour as conditioned responses. Imminence to the US is an important one, but so too are the physical properties of the CS. Holland (Holland, 1979; 1980a,b among others) has shown this convincingly for both appetitive and aversive Pavlovian conditioning. There are CS generated behaviours that can be increased across conditioning and there are US generated behaviours that can also increase across conditioning. So, any dichotomy between freezing and escape is not absolute if one is CS generated (escape) and the other is US generated (freezing).

I think the authors could embrace this complexity a little more. There are multiple determinants of the form of the CR. As conducted here, the physical properties of the CSs (frequency and sound pressure levels) are important, But, Dong et al., found the opposite. They showed robust escape responses to a tone when it was a proximal CS and freezing to the white noise when it was a distal CS. This difference is never really reconciled. Nor are the present data reconciled with the findings of Fadok et al. showing that the white noise CS in their experiments did not elicit escape in the absence of shock. This contrasts with Experiment 2/Figure 2 here.

2) Is there evidence for Pavlovian conditioning to the auditory CSs?

If the authors are seeking to argue that behaviour to the auditory CSs are conditioned responses, then they need to show evidence for conditioning.

I was struggling to understand the evidence the authors were invoking for Pavlovian conditioning to the CSs. The authors have three groups: two paired groups that receive a serial compound comprised of two auditory CS, white noise and tone, followed by a US. The groups differ simply in the order of the two CSs. The third group received tone then white noise with a 60 s interval between the offset of the CS and delivery of the US. This is an "unpaired" group. The inclusion of a control is to be commended. It is a conservative control, because it is really a trace conditioning, rather than unpaired, control. Regardless, from this kind of design, the evidence for conditioning to the CSs would be to show that freezing and escape responses were significantly greater in the two paired groups compared to the unpaired control (i.e., a 3-way ANOVA with a 3-way interaction driven by more responding in the two paired groups than the unpaired group). It is hard to tell from Figure 1 if this will come out. I suspect it will not for freezing but it may for average motion and escape score (see next point). I encourage the authors to consider this analysis. They need to persuade readers that they are studying learning before they persuade readers that CS salience determines topography of defensive behavior as conditioned responses. This same principle applies to Figure 1—figure supplement 1, as well as Figure 4. In fact, I really could not see any evidence that conditioning to the CSs occurred (if it is defined relative to the control unpaired group) in many key experiments in this paper.

A different solution could be to report pre-CS levels of freezing/escape for *each* CS presentation and show significant increases in freezing/escape during CS presentations relative to each 10s pre-CS period. Less ideal would be to show that there is more freezing/escape across CS presentations than in the pre-CS/baseline period.

The reason this is important is simply that one could interpret the data as showing contextual fear conditioning in each group upon which different unconditioned responses to the CSs are superimposed.

3) Is salience, imminence, or both important?

A key conclusion from this paper is that salience determines the topography of conditioned fear responses when these auditory stimuli are used. For example, in Experiment 1 the key conclusion here is white noise CS elicits escape behavior regardless of whether it is proximal or distal to the US whereas tone CS elicits freezing and not escape. In Experiment 4 a similar conclusion applies to the 12kHz vs.3 kHz tone. This seems reasonable based on inspection of the figures. To be sure, there is rarely if any escape to the tone CS in Experiment 1/Figure 1. However, I am not sure it is the complete answer. The related, critical question is whether the topography of defensive behavior to the CSs depends significantly on their temporal relation to the US? In Experiment 1, does the white noise CS elicit *more* escape responses when it is proximal rather than distal to the US? This analysis, like the evidence for conditioning described above, simply requires a 3-way ANOVA, in this case comparing behaviors between the two Paired groups, and specifically testing a 3-way interaction (G1 vs. G2 or G4 vs. G5 x CS [noise vs. tone] or [3kHz vs. 12 kHz tone] x trials [1 – 10]). This would test whether the difference between each response to the two kinds of CSs is or is not significantly greater between the two paired groups. If identity of the CS, not its temporal order, is important then this interaction should not be significant. If temporal order is important, then this interaction will be significant. For example, is the difference between the motion (Figure 1J) or escape scores (Figure 1M) for white noise versus tone significantly greater in Group 1 versus Group 2? If this interaction is significant, it lends support to the claim that temporal imminence matters, at least in part. If it is not, it lends support to the claim that temporal imminence does not matter.

I realize the authors want to focus on the lack of any real escape responses to the tone CS in Experiment 1 or the 3 KHz tone in Experiment 4, and this is important as well as obvious from the data. However, the data appear more nuanced. The data as presented do appear to suggest that the extent of escape responses to the white noise are determined, at least in part, by the temporal relation of the CS to the US (imminence) as well on the specific physical properties of the CS (see point 1).

I am not recommending any additional data collection. The manuscript is interesting, challenging, and also instructive. However, I think:

1) Further analyses are needed, specifically the 3-way ANOVAs described above. One set of 3 ANOVAs asking whether each behavior is different across trials for the control vs. two paired groups. A second set asking whether there is a difference between the two paired groups for each behavior. The authors strategy of analyzing the groups separately undermines their key conclusions.

2) deeper consideration of the role of CS and US generated behaviors as conditioned responses is warranted as is further attempts to reconcile these findings with Fadok et al., and Dong et al.

*Reviewer #3:*

This study examines the recent claim that conditional stimuli (CSs) presented proximal to footshock elicit escape responses while distal CSs induce freezing behavior. The authors make a convincing case that escape responses are controlled by the salience of the CS rather than its proximity to shock. Previous work had used white noise as the cue that was presented immediately prior to shock. The current experiments counterbalanced this cue with a pure tone and found that white noise produced bursting whether it was located proximal or distal to footshock. In contrast, the pure tone induced freezing even when it was presented immediately prior to shock. However, if the salience of the pure tone was enhanced by increasing its intensity, then it was able to drive some escape behaviors. Therefore, in contrast to previous claims, defensive behaviors elicited by auditory CSs (in mice) are primarily controlled by stimulus salience. These results have important implications for studies that use serial compound cues to study proximal and distal threat responses.

One thing I would like the authors to address is that there appears to be more bursting to the white noise stimulus when it is presented as CS2 compared to CS1. This suggests there is an interaction between stimulus salience and proximity to threat (i.e. white noise produces more bursting than a pure tone and the size of this response is larger when the stimulus occurs proximal to footshock).

A second issue is the amount of nonassociative CRs that occurs in the unpaired groups. It is possible that with the strong shocks that are used, mice are able to associate the CS with the US. That is, the unpaired procedure is actually a trace conditioning procedure. Have the authors tried using a longer gap between the CS and shock? Or presenting the CS and shock on different training days? Does this reduce the amount of nonassociative responding?

---

## [Author Response]

Essential revisions:Reviewer #1:1) There are two main areas I think could be made clearer and perhaps softened a bit. The first concerns the relationship between these findings and the general idea that behaviors to cues can differ based on temporal factors, salience, or even cue modality. I do not know the SCS field particularly, but even without a compound cue, it is well documented that both unconditioned and conditioned behaviors differ across time, particularly during a long (10, 20, 30s) cue. These behaviors and their relationships can also differ based on the amount of training, and the density of reward I believe. So, it seems to me, that there are many factors that can explain differential responding of the sort that seems to be dogmatically-highlighted in the SCS literature. I wonder if a more general introduction that acknowledges some of this complexity might be considered, versus what seems to be a dichotomy in the current Introduction?

This is an important point, and a good suggestion. We have:

a) Added a new paragraph (Introduction) that acknowledges the complexity of behavioral responses to cue stimuli, along with references to classic work which demonstrated this.

b) Moved the section highlighting the difference between our results and those of Dong et al., to the Discussion section.

2) Related to this, I think the authors should soften their conclusions a bit. At present, it seems to me that they are saying the SCS effect is solely due to salience differences. While this may be the case, I think this is conclusion goes beyond what is necessary and the data.

Agreed. We have altered the language used to summarize our findings to avoid giving the impression that we believe salience is the sole explanation of behavioral responding in the SCS paradigm. Specifically, we:

a) State that stimulus salience is the ‘primary determinant’ (Abstract), or ‘major factor’ (Introduction) determining behaviour.

b) Conclude that ‘audio frequency properties strongly influence defensive behaviors elicited by SCS’ conditioned stimuli (Discussion section), and that salience is the “primary means by which mice assess imminence…” (Discussion section).

c) Added a new paragraph to acknowledge “it remains possible that temporal factors make some contribution to defensive responding in SCS conditioning” (Discussion section).

Reviewer #2:3) It is well established that multiple factors determine the topography of behaviour as conditioned responses. Imminence to the US is an important one, but so too are the physical properties of the CS. Holland (Holland, 1979; 1980a,b among others) has shown this convincingly for both appetitive and aversive Pavlovian conditioning. There are CS generated behaviours that can be increased across conditioning and there are US generated behaviours that can also increase across conditioning. So, any dichotomy between freezing and escape is not absolute if one is CS generated (escape) and the other is US generated (freezing). I think the authors could embrace this complexity a little more.

Agreed, this was also suggested by reviewer #1; please see our response on this point above (issues #1 and 2).

4) There are multiple determinants of the form of the CR. As conducted here, the physical properties of the CSs (frequency and sound pressure levels) are important, But, Dong et al., found the opposite. They showed robust escape responses to a tone when it was a proximal CS and freezing to the white noise when it was a distal CS. This difference is never really reconciled.

We made multiple attempts to contact the corresponding senior author of the Dong et al. study via e-mail in order to obtain additional details of their experiments that could help reconcile the differences with our results. Unfortunately, they did not respond to our e-mails, and so our ability to explain the discrepant results is limited to scrutinizing methodological details reported in their paper, which appear essentially the same as those that we employed. As the Dong et al. study did not specify the particular mouse substrain used in their experiments, we have proposed this as a potential explanation, and suggest that future work would be required to determine if different mouse substrains use distinct processes to determine threat imminence (Discussion section).

5) Nor are the present data reconciled with the findings of Fadok et al. showing that the white noise CS in their experiments did not elicit escape in the absence of shock. This contrasts with Experiment 2/Figure 2 here.

The results in Experiment 2/Figure 2 of our manuscript were done in *unconditioned* mice and did not examine escape behavior, but rather physiological (pupil dilation) and simple locomotor responses (movement of head-fixed animals on a running wheel) to tone and white noise stimuli to which the animals had not been previously exposed. Therefore, our data are not directly comparable to the experiments that Fadok et al. performed on conditioned animals. To clarify this for readers, we modified the text to read “simple locomotor responses on the running wheel” in the section detailing the results of Figure 2 (Results section).

This being said, it is important to note that supplemental figure 1 of the Fadok et al., study did in fact provide evidence that white noise is more salient and/or threatening to mice than tone stimuli. Specifically, when they performed conventional fear conditioning to a simple CS composed of just a single auditory stimulus, significantly more flight behavior was evoked by a white noise CS than a tone CS (Fadok et al., 2017, Extended Data Figure 1F). We have added explicit mention of this point to the manuscript, as well as other evidence that white noise may be uniquely salient if not threatening to mice (Discussion section).

6) If the authors are seeking to argue that behaviour to the auditory CSs are conditioned responses, then they need to show evidence for conditioning.I was struggling to understand the evidence the authors were invoking for Pavlovian conditioning to the CSs. The authors have three groups: two paired groups that receive a serial compound comprised of two auditory CS, white noise and tone, followed by a US. The groups differ simply in the order of the two CSs. The third group received tone then white noise with a 60 s interval between the offset of the CS and delivery of the US. This is an "unpaired" group. The inclusion of a control is to be commended. It is a conservative control, because it is really a trace conditioning, rather than unpaired, control. Regardless, from this kind of design, the evidence for conditioning to the CSs would be to show that freezing and escape responses were significantly greater in the two paired groups compared to the unpaired control (i.e., a 3-way ANOVA with a 3-way interaction driven by more responding in the two paired groups than the unpaired group).

Thank you for raising this important point; we have addressed it as follows:

a) TN-WN SCS conditioning data (Figure 1) was analyzed using 3-way ANOVA:

i) Freezing to TN was higher in paired Group 1 (G1) than unpaired Group 3 (G3) (3-Way ANOVA on Day 2, G1 vs. G3, Main Effect of Stimulus (F(1,23) = 429.5, p<0.0001), Main Effect of Trial (F(4,92) = 5.083, p=0.001), Group X Stimulus Interaction, (F(1,23) = 27.51, p<0.0001); Follow-Up Two-Way RM ANOVA for freezing just to the tone stimulus, Main Effect of Trial (F(3.2, 75.9) = 3.79, p<0.05), Main Effect of Group (F(1,23) = 6.41, p<0.05)).

ii) Motion to WN was higher in G1 than G3 (3-Way ANOVA on Day 2, G1 vs. G3 Activity, Main Effect of Stimulus (F(1,23) = 69.89, p<0.0001), Main Effect of Group (F(1,23) = 11.75, p<0.01), Group X Stimulus Interaction (F(1,23) = 19.77, p<0.001); Follow-up Two-Way RM ANOVA for activity just to WN stimulus, Main Effect of Group (F(1,23) = 15.79, p<0.001)).

iii) Escape scores during WN were higher in G1 than G3 (3-Way ANOVA on Day 2, G1 vs. G3 Escape Score, Main Effect of Stimulus (F(1,23) = 67.85, p<0.0001), Main Effect of Group (F(1,23) = 15.98, p<0.001), Group X Stimulus Interaction (F(1,23) = 20.41, p<0.001); Follow-up Two-Way RM ANOVA for escape score just to WN, Main Effect of Group (F(1,23) = 18.25, p<0.001)).

iv) Paired Group 2 (G2) displayed significantly different behavioral responses to the two CS stimuli across all metrics (2-Way ANOVA results already reported) and in the same direction as G1 (i.e. more freezing during TN, more activity and escape during WN). However, G2 did not show significantly different magnitude of these responses compared with G3 (3-Way ANOVA on Day 2, G2 vs G3, no group differences or interactions for freezing, motion, or escape score). We hypothesize that this may be due to impaired learning when using a high-to-low salience SCS (see more on this issue below in point ‘3’).

b) To directly demonstrate an acute conditioned freezing response to the TN stimulus, we also performed new experiments in which a tone test in a novel context was performed following SCS conditioning of mice trained on group1 (paired TN-WN-US) or group 3 (gap TN-WN-gap-US) protocols. Whereas mice in the gap group (G3 protocol) did not show significantly increased freezing between baseline and tone onset (p>0.05), paired (G1 protocol) mice exhibited robust freezing upon tone onset (p<0.001)(Two-Way RM ANOVA, Main Effect of Stimulus (F(1,13) = 19.98, p<0.001), Stimulus X Group Interaction (F(1,13) = 5.492, p<0.05), Sidak’s comparisons to determine which group drives the Main Effect of Stimulus). This data has been added to a new figure supplement (Figure 1—figure supplement 1).

c) Additional evidence that conditioning occurred to both TN and WN stimuli comes from the experiments detailed in Figure 3:

i) Mice in paired G1 showed significantly higher motion and escape scores than gap G3 mice to TN stimuli in the tone SPL step test (Figure 3F-I).

ii) Mice in paired G1 showed significantly higher freezing than gap group G3 in response to WN stimuli at low SPL, and G1>G3 for escape score in response to WN at high SPL (Figure 3M-O).

iii) To clarify this for readers, we modified the text to indicate that “group 1 responses are at least in part influenced by perceived threat levels which are a function of conditioned fear” (Results section).

d) For the two-tone SCS experiments (Figure 4):

i) Motion to 12 kHz tone was higher in G4 than G6: 2-Way RM ANOVA, G4 vs. G6, Motion to 12 kHz tone: Main Effect of Trial (F(2.5, 45.7) = 3.31, p<0.05), Main Effect of Group (F(1,18) = 8.64, p<0.01).

ii) Escape score to 12 kHz tone was higher in G4 than G6: 2-Way RM ANOVA, G4 vs. G6; Main Effect of Trial (F(2.7, 48.0) = 4.36, p<0.05), Main Effect of Group (F(1,18) = 10.1, p<0.01).

iii) Escape score to 12 kHz tone was higher in G5 than G6: 2-Way RM ANOVA, G5 vs G6, Escape score to 12 kHz tone: Main Effect of Group (F(1,18) = 4.49, p<0.05).

iv) Freezing to 3 kHz tone was higher in G4 than G6 in a tone test: Though groups did not differ in freezing behavior to the 3 kHz tone during conditioning, this difference was revealed in novel context tone test (Figure 4N-P).

In sum, these analyses indicate that conditioning to the auditory stimuli occurred in groups 1 and 4, and that the active and passive behaviors elicited by these stimuli are at least in part a function of Pavlovian conditioning. Moreover, reversing the order of TN and WN (or 3 and 12 kHz tones) during conditioning does not reverse the behaviors elicited by these stimuli. Details of these analyses have been added to the text (Results section), a *new figure* has been added (Figure 1—figure supplement 1), and the specific statistical tests performed appended to the source data sheets.

7) The related, critical question is whether the topography of defensive behavior to the CSs depends significantly on their temporal relation to the US? In Experiment 1, does the white noise CS elicit more escape responses when it is proximal rather than distal to the US?

Reversing order of SCS component stimuli did not reverse behaviors elicited (i.e. both G1 and G2 froze more during TN presentations and exhibited higher motion and escape in response to WN). Therefore, temporal relationship of a CS to the US is not the key factor determining whether mice execute an active or passive behavior. However, the data do clearly indicate that SCS order has an effect of the *magnitude* of behavioral responding; this was revealed in the 3-way ANOVA analyses, which showed that G2 (paired) active and passive behaviors are not significantly different from G3 (gap). One explanation for these data is that the reversed WN-TN SCS may impair learning, and we have added a new section that explicitly highlights these quantitative (but not qualitative) differences between G1 and G2 behavior, as well as propose potential explanations for why a high-to-low salience SCS might impair learning (Discussion section). In addition, although salience appears to be the major explanation for differential responding in SCS conditioning, we acknowledge that some contribution of temporal association cannot be excluded, and so added a new statement to this effect (Discussion section).

Reviewer #3:8) One thing I would like the authors to address is that there appears to be more bursting to the white noise stimulus when it is presented as CS2 compared to CS1. This suggests there is an interaction between stimulus salience and proximity to threat (i.e. white noise produces more bursting than a pure tone and the size of this response is larger when the stimulus occurs proximal to footshock).

Yes, this is an important observation also noted by reviewer #2. Please see our response on this point above (issue #7).

9) A second issue is the amount of nonassociative CRs that occurs in the unpaired groups. It is possible that with the strong shocks that are used, mice are able to associate the CS with the US. That is, the unpaired procedure is actually a trace conditioning procedure. Have the authors tried using a longer gap between the CS and shock? Or presenting the CS and shock on different training days? Does this reduce the amount of nonassociative responding?

Agreed, this is an important point which we have addressed as follows:

a) We changed the name of Groups 3 and 6 from ‘unpaired’ to ‘gap’ to indicate that these groups are not fully ‘unpaired’, and to reflect the possibility that some CS-US association may have developed despite the 60s gap between the SCS and US.

b) We added a new paragraph to explicitly acknowledge that the gap group may have undergone trace conditioning, and that future work would be needed to determine the extent to which the behaviors exhibited by the gap groups are nonassociative (Discussion section).